# 5th generation vs 4th generation troponin T in predicting major adverse cardiovascular events and all-cause mortality in patients hospitalized for non-cardiac indications: A cohort study

**Vedant Gupta**[1¤]*, **Marc Paranzino**[1], **Talal Alnabelsi**[1], **Karam Ayoub**[1], **Joshua Eason**[1], **Andin Mullis**[1], **John R. Kotter**[1], **Andrew Parks**[2], **Levi May**[2], **Sethabhisha Nerusu**[3], **Chen Dai**[3], **Daniel Cleland**[3], **Steve Wah Leung**[1], **Vincent Leigh Sorrell**[1]

**1** Gill Heart and Vascular Institute, University of Kentucky, Lexington, Kentucky, United States of America,
**2** Department of Internal Medicine, University of Kentucky, Lexington, Kentucky, United States of America,
**3** Performance Analytics Center of Excellence, University of Kentucky, Lexington, Kentucky, United States of America

¤ Current address: Gill Heart and Vascular Institute, Lexington, Kentucky, United States of America
* Vedant.gupta@uky.edu

**Data Availability Statement:** All relevant data are within the manuscript and its Supporting Information files.

## Abstract

### Objective

The frequency and implications of an elevated cardiac troponin (4th or 5th generation TnT) in patients outside of the emergency department or presenting with non-cardiac conditions is unclear.

### Methods

Consecutive patients aged 18 years or older admitted for a primary non-cardiac condition who had the 4th generation TnT drawn had the 5th generation TnT run on the residual blood sample. Primary and secondary outcomes were all-cause mortality (ACM) and major adverse cardiovascular events (MACE) respectively at 1 year.

### Results

918 patients were included (mean age 59.8 years, 55% male) in the cohort. 69% had elevated 5th generation TnT while 46% had elevated 4th generation TnT. 5th generation TnT was more sensitive and less specific than 4th generation TnT in predicting both ACM and MACE. The sensitivities for the 5th generation TnT assay were 85% for ACM and 90% for MACE rates, compared to 65% and 70% respectively for the 4th generation assay. 5th generation TnT positive patients that were missed by 4th generation TnT had a higher risk of ACM (27.5%) than patients with both assays negative (27.5% vs 11.1%, p<0.001), but lower than patients who had both assay positive (42.1%). MACE rates were not better stratified using the 5th generation TnT assay.

**Funding:** The project described was supported by the University of Kentucky Center for Health Services Research Data, Analytics, and Statistical Core. Roche Diagnostics provided the reagents for the 5th generation Troponin T free of cost. No additional renumeration was provided. The funders had no role in study design, data collection and analysis, decision to publish, or preparation of the manuscript.

**Competing interests:** The commercial funder, Roche Diagnostics provided the reagents for the 5th generation Troponin T assay free of cost. No additional renumeration was provided to any authors. Roche Diagnostics was not involved in study design, data collection, analysis, or manuscript writing. This does not alter our adherence to PLOS ONE policies on sharing data and materials. Furthermore, there are no competing interests with Roche Diagnostics or other relevant entities for any of the authors with regard to employment, consultancy, patents, products in development, marketed products, etc.

## Conclusions

In patients admitted for a non-cardiac condition, 5th generation TnT is more sensitive although less specific in predicting MACE and ACM. 5th generation TnT identifies an intermediate risk group for ACM previously missed with the 4th generation assay.

## Introduction

Cardiac troponin is the recommended biomarker for the diagnosis of acute coronary syndrome (ACS), and its role in the rapid detection of ACS is well established [1, 2]. The transition from contemporary sensitivity to the high-sensitivity cardiac troponin has increased the ability to reliably rule out patients for suspected ACS at the expense of specificity [2–5]. Consequently, a large patient population has been identified with increased troponin values above the 99th percentile who do not meet the 4th Universal Definition of Acute Myocardial Infarction [1]. Recent studies showed worse cardiovascular outcomes in undifferentiated emergency department (ED) patients with elevated high sensitivity troponin values and both contemporary sensitivity and high sensitivity troponin values are associated with outcomes in the ED chest pain patients [6, 7]. More recent studies have tried to assess the implications of type 2 myocardial infarction, acute myocardial injury and chronic myocardial injury, but these are often limited to patients with biomarkers drawn in the ED and include patients presenting with a primary cardiac complaint (albeit non-ACS) [8–10]. However, considerably less is known regarding the implication of increased troponin in patients hospitalized for non-ACS conditions.

In a related note, cardiovascular risk is elevated in many non-cardiac conditions and troponin is known to be associated with all-cause mortality (ACM) in these non-ACS clinical populations [11–14]. A recent, large retrospective trial demonstrated that elevation in troponin is associated with increased major adverse events in the absence of coronary artery disease, clinical heart failure or renal dysfunction [12]. To date, no studies have directly compared contemporary and high sensitivity troponin values in hospitalized patients with a non-ACS presentation. Many guideline documents recommend the routine assessment of troponin for non-ACS presentation despite incompletely established risk associations and with a limited understanding of the downstream implications [15–17]. This study aims to assess the predictive nature of cardiac troponin while comparing different generations of troponin T assays.

## Methods

### Study population

This study included patients 18 years of age and older admitted to our quaternary care center (University of Kentucky Medical Center) for a primary non-cardiac condition who had a contemporary sensitivity troponin assay ordered between January 2017 to October 2017. The clinical laboratory identified patients based on the contemporary troponin assay being run from all locations excluding our cardiovascular care units (CCU), primary cardiovascular floors, Children's Hospital, floors that housed adolescent or pediatric patients, and emergency department. All patients had study procedures run on residual blood samples within 30 minutes after it was deemed that there was no additional clinical need for the sample. All charts were subsequently reviewed for appropriateness of inclusion. Patients were excluded from the final analysis if they were admitted to a primary cardiovascular service despite being on a traditionally

non-cardiac service, presented with a primary cardiac complaint (chest pain or equivalent, heart failure exacerbation, or arrhythmia) even if on a non-cardiac service, or presenting within 7 days of a primary PCI or 30 days from an electrophysiologic or cardiothoracic procedure.

The study was designed to be a matched comparison allowing for each study participant to serve as their own control as they had both the high sensitivity and contemporary sensitivity assay run. The comparison was to the predictive ability of each assay in predicting all-cause mortality and CV events. Given there was no difference in event rates between the control and study arm, a traditional sample size calculation was deferred. However, based on a previous study of septic patients done at our institution, we anticipated a 30% all-cause mortality rate and a 7% major adverse cardiovascular event rate in our study population with 3000 samples requested (3 samples per participant and 1000 participants included in the study) [18].

The Institutional Review Board (IRB) approved the protocol and waived need for informed consent given no patient contact was made by study personnel and the blood sample used for study procedures was residual blood that would have otherwise been discarded. Roche Diagnostics provided the reagents prior to approval of the high sensitivity troponin assay by the Food and Drug Administration as a part of an Investigator Initiated Research Agreeement.

## Study procedures

All patients having a clinically indicated 4th generation Elecsys Troponin T immunoassay (Roche Diagnostics, Germany) drawn reflexively had the 5th generation Elecsys Troponin T immunoassay (Roche Diagnostics, Germany) run on the residual blood sample. Local validation and precision studies were done for both assays to identify 99th percentile cutoffs. For the 5th generation troponin T assay (5th generation TnT), gender specific cutoffs were used as per the recommendations from Roche. Gender specific cutoffs were determined by running assays on 300 healthy male and 300 healthy female volunteers. The 99th percentile from this normative data set was identified, and then compared to those reported in the literature. Abnormal values for the 5th generation TnT assay were >13 ng/L for females and >18 ng/L for males. Abnormal values for the 4th generation troponin T assay (4th generation TnT) were ≥0.01 ng/mL. Troponins were deemed elevated if the peak value based on the assay specific cutoffs were seen at any point in the hospitalization (not just index troponin). Given it was a new assay not yet approved by the Food and Drug Administration, the 5th generation TnT was not reported out clinically.

All clinical charts included were extensively reviewed to identify reason for admission, clinical symptom or indication for troponin being drawn, ECG findings, clinical risk factors, and additional cardiovascular testing results. ECG findings were reported based on the clinical read of a board-certified cardiologist at the time care. A random sample of 100 charts were de-identified and reviewed by a board-certified cardiologist (VG) to ensure consistency and accuracy of documentation.

## Outcomes

The primary outcome for this study was all-cause mortality (ACM) at 1 year. The secondary outcome was major adverse cardiovascular events (MACE) at 1 year. MACE included cardiovascular death, acute myocardial infarction (MI) or revascularization (PCI or CABG). Cardiovascular death was defined as fatal ventricular arrhythmia or cardiogenic shock as a primary or significant secondary contributor (defined as mixed shock requiring inotropes and either significant LV dysfunction or low mixed venous oxygen saturation). MI was defined using the 4th Universal Definition of Myocardial Infarction [1]. All outcomes were identified through

review of the electronic health record. To minimize missing data, the Kentucky Health Information Exchange was also accessed for potential admissions or testing at other participating hospitals. The Kentucky Health Information Exchange is a consortium of over 1500 medical centers, independent providers and pharmacies within the state to optimize data sharing. All major cardiovascular events, including CV death were adjudicated by 2 board certified cardiologist who were blinded to their 5[th] generation TnT levels (JK and VG).

After directly comparing 4[th] generation versus 5[th] generation TnT, the cohort was then divided into three groups: Group 1 was concordant negative (both 4[th] vs 5[th] generation TnT negative), Group 2 was discordant (4[th] generation TnT negative, 5[th] generation TnT positive), and Group 3 was concordant positive (both 4[th] and 5[th] generation TnT positive). This was to identify the additive impact of the transition from the 4[th] generation to the 5[th] generation assay in predicting ACM and MACE at 1 year. No patients had a positive 4[th] generation TnT and a negative 5[th] generation TnT.

## Statistical analysis

All categorical variables were reported in percentages and all continuous variables are reported in means and standard deviations for normally distributed variables and medians and interquartile ranges for non-normally distributed variables. Sensitivities, specificities and accuracies were calculated for the 5[th] generation TnT assay and the 4[th] generation TnT assay for ACM and MACE at 1 year. Then, receiver operator curves were generated for each test. Finally, the ACM and MACE rates in the prespecified groups (concordant negative, discordant, and concordant positive groups) were compared using Chi square analysis. SPSS v26 (IBM, Texas, USA) was used for all analysis. A p value of <0.05 was used to assess statistical significance.

## Results

### Study population

A total of 927 were prospectively screened as having a 4[th] generation TnT ordered from the inpatient, non-cardiovascular units. Nine patients were excluded due to a primary cardiovascular reason for admission or admission to a cardiology service. No patients were excluded for a recent cardiovascular procedure. A total of 918 patients were included in the final analysis (S1 Fig). The mean age of the study population was 59.8 years with 45% female. This population was noted to have significant cardiovascular risk factors, including hypertension (72%), hyperlipidemia (52%), diabetes (24%), and known coronary artery disease (28%). Full baseline demographic data is available in Table 1.

Patients primary reason for admission was also documented based on what was known at the time of the troponin draw. Of the 918 patients included, the most common reason for admission was acute respiratory failure (23.3%), acute or chronic stroke (23.3%), sepsis (22.2%), hematology/oncology related illness (9.9%), trauma (7.6%), gastrointestinal bleeding (6.0%), and pneumonia (5.7%). A non-cardiovascular operation was performed in 15.4% of patients during that hospitalization prior to the troponin being drawn. Many patients had multiple reasons for admission. Full breakdown of reason for admission is seen in Fig 1.

### Clinical symptom triggering workup and ECG results

Clinical symptoms and indications that prompted ordering of the 4[th] generation TnT assay was assessed (Table 2). Of the 918 patients, 43.9% had no reported cardiovascular symptoms that triggered troponin collection. Of the remaining patients, 17.8% reported dyspnea while 17.7% patients experienced preceding chest pain triggering ordering of troponin. Other cardiovascular

**Table 1. Baseline demographics.**

|  | Frequency (N = 918) | Percent |
|---|---|---|
| **Age, yrs.** | 59.8 +/- 15.6 years | |
| **Male** | 506 | 55.1% |
| **Female** | 412 | 44.9% |
| **BMI** | 29.4 +/- 8.3 | |
| **Tobacco** | | |
| Never smoker | 345 | 37.6% |
| Former smoker | 223 | 24.3% |
| Current smoker | 290 | 31.6% |
| Smoking status not known | 60 | 6.5% |
| **Previous Medical Conditions** | | |
| Hypertension | 658 | 71.7% |
| Hyperlipidemia | 479 | 52.2% |
| Diabetes | 340 | 37.0% |
| Coronary Artery Disease | 254 | 27.7% |
| Cerebrovascular Disease | 224 | 24.4% |
| Chronic Kidney Disease | 214 | 23.3% |
| Peripheral Artery Disease | 58 | 6.3% |

symptoms included hypotension (9.1%), arrythmia (5.8%) or rhythm change (5.3%). There were also 75 patients (8.2%) who reported non-cardiac symptoms at the time of troponin collection.

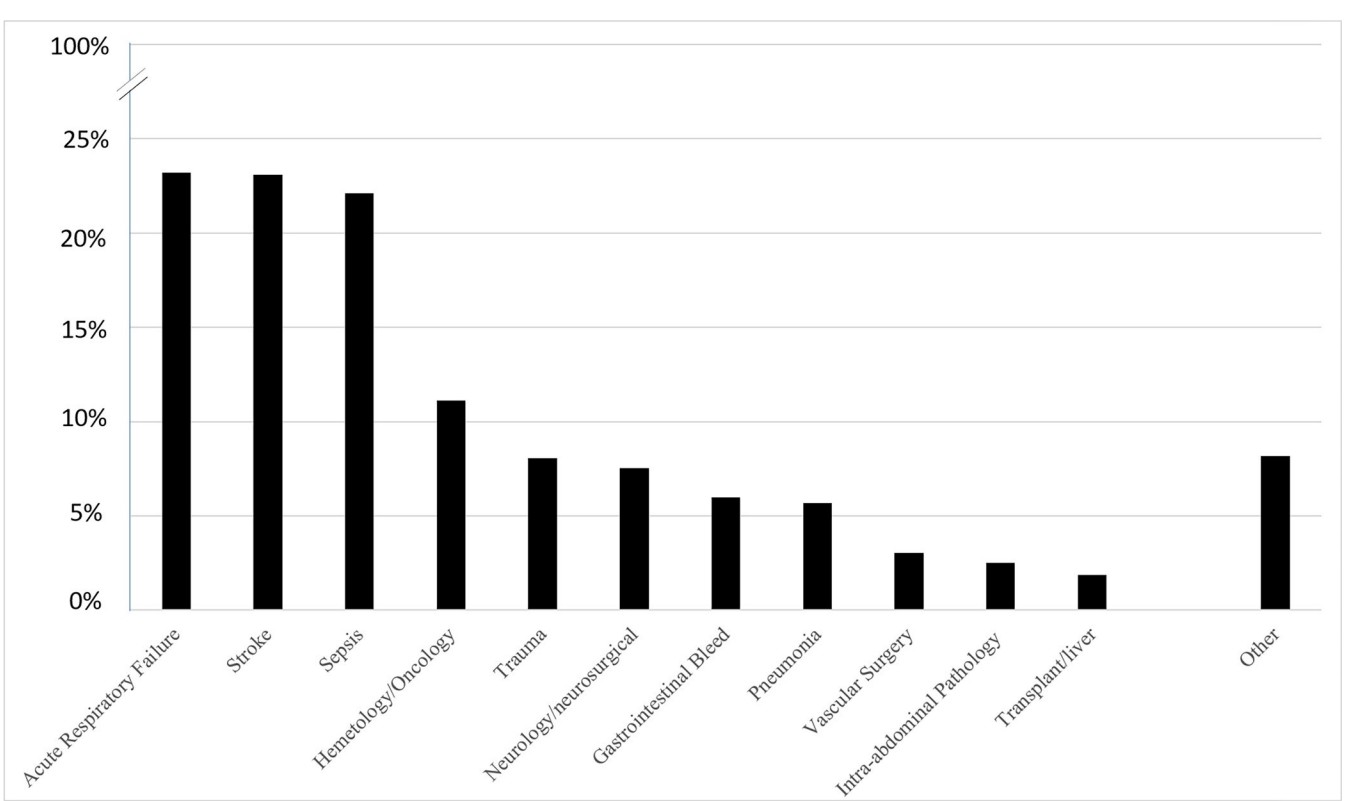

**Fig 1. Reason for index hospitalization.**

**Table 2. Symptoms/indications for troponin and ECG findings.**

| Symptoms triggering ordering of troponin | | |
|---|---|---|
| Shortness of air | 163 | 17.8% |
| Chest Pain | 162 | 17.6% |
| Hypotension | 83 | 9.0% |
| Arrythmia | 53 | 5.8% |
| Rhythm Change | 51 | 5.6% |
| Other | 75 | 8.2% |
| No discrete symptoms documented | 403 | 43.9% |
| **Electrocardiogram Obtained** | 853 | 92.9% |
| **Electrocardiogram Results** | | |
| No ST deviation | 374 | 43.8% |
| Non-Specific ST/T Wave Abnormality | 333 | 39.0% |
| T-wave Inversion | 14 | 1.6% |
| ST Elevation | 13 | 1.5% |
| ST Depression | 4 | 0.5% |
| Conduction Abnormality | 159 | 18.6% |
| Other | 173 | 20.3% |

No discrete symptoms were documented in over 43% of patients, most commonly as a part of a routine order set for stroke patients in concordance with guideline documents [17].

ECGs were ordered in 93% of patients in whom a troponin was drawn. Over 50% of the patients did not have any appreciable ECG abnormalities. Approximately 40% of patients had non-specific ST/T wave abnormalities, and <10% had >1 mm ST depression or elevation on their ECG.

## Outcomes

Median peak 5[th] generation TnT values were 34 ng/L (IQR of 13–98 ng/L) while the median peak 4[th] generation TnT values were 0.01 ng/mL (IQR of 0.01–0.071 ng/mL). Using the aforementioned cutoffs for normal values, 69% (n = 635) of the study population had elevated 5[th] generation TnT levels while 46% (n = 420) of the study population had elevated 4[th] generation TnT levels. Correlation between the 5[th] generation and 4[th] generation TnT results focusing on the lower levels are shown in S2 Fig.

The 5[th] generation TnT assay was more sensitive, but less specific for predicting ACM and MACE than the 4[th] generation TnT assay. The sensitivities for the 5[th] generation TnT assay were 85% for ACM and 90% for MACE rates, compared to 65% and 70% respectively for the 4[th] generation assay. The specificities for the 5[th] generation TnT assay were 45% for ACM and 35% for MACE rates, compared to 55% and 60% respectively for the 4[th] generation TnT assay (S1 and S2 Tables). There was no difference in overall accuracy (Fig 2).

To classify, 286 patients were concordant negative (4[th] generation TnT -/5[th] generation TnT -), 215 patients were discordant (4[th] generation TnT -/5[th] generation TnT +), and 420 patients were concordant positive (4[th] generation TnT +/5[th] generation TnT +). ACM and MACE for the concordant negative was noted to be 11.1% and 1.0% respectively. Compared to the concordant negative patients, the discordant patients had higher ACM rates at 27.5% (p < 0.001) with no significant increase in MACE rates at 1.7% (p = 0.48). Finally, the concordant positive patients had the highest rates of ACM at 42.1% (p < 0.001) and MACE rates at 5.9% (p < 0.0001) (Fig 3). MACE rates were distributed amongst all the different reasons for admission with no statistically significant difference between the different categories (S3 Fig).

A

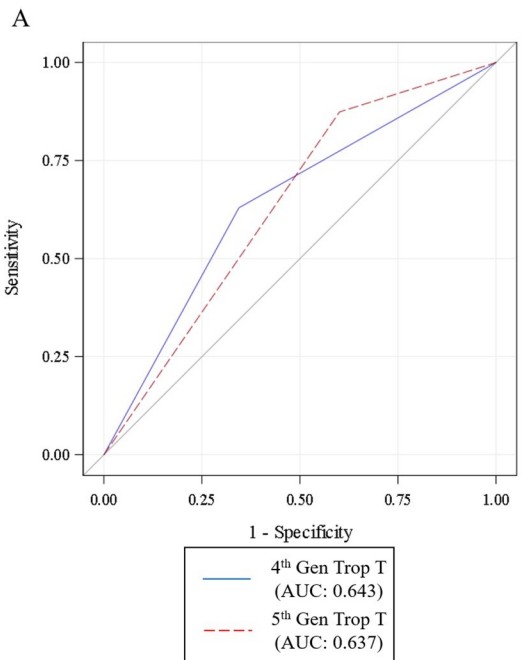

B

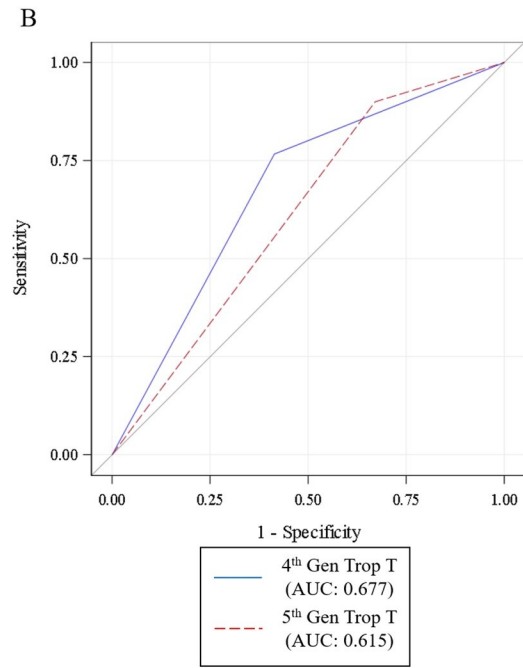

**Fig 2.** ROC curves for 4th generation troponin T and 5th generation troponin T for all-cause mortality (A) and MACE (B). ROC curves for both troponin assays with no significant difference in the areas under the curve (AUC) both for all-cause mortality and MACE (p values >0.05).

## Discussion

The primary findings highlighted by this investigation include the following: 1) cardiovascular risk is elevated compared to general populations, but not as high as amongst those with primary cardiovascular conditions reported in the literature; 2) 5th generation TnT assays are more sensitive and identify a previously unidentified cohort of patients at risk for ACM but not for MACE; and 3) cardiac troponin alone (either 4th generation or 5th generation TnT) are likely not adequate as a diagnostic tool in isolation, but need to be integrated into a more comprehensive risk stratification schema.

The introduction of higher sensitivity troponin assays has largely focused on subgroups routinely assessed in the emergency department, either as those with suspected ACS or other non-differentiated patients. However, troponin is routinely assessed in the hospitalized patients for various reasons, including as a part of routine order sets, non-specific symptoms, or a decompensating patient. In addition, there are more studies regularly that have emphasized cardiovascular risk in these non-ACS hospitalized patients [10–12]. Understanding the connection (and therefore the interpretation) between cardiac biomarkers and cardiovascular risk is essential in these hospitalized patients, especially given that >2/3rd of hospitalized patients in this cohort have elevated 5th generation TnT values, an absolute increase in 20–25% compared to the 4th generation TnT.

Cardiovascular risk has recently been assessed broadly and with variable reported risk amongst hospitalized patients [17–19]. This wide variation is often related to definitions and methodology used. Coded data often overestimates the risk of cardiovascular events, and often will yield complication rates as high as 25% at 1 year [20, 21]. Our study was extremely stringent in outcome assessment, excluding all patients who were identified as having an ACS during their initial presentation, adhering to the 4th Universal Definition of Myocardial Infarction and requiring agreement between 2 cardiologists for cardiovascular outcomes [1]. With these more stringent criteria, the MACE rate was higher at 1 year than a non-hospitalized cohort,

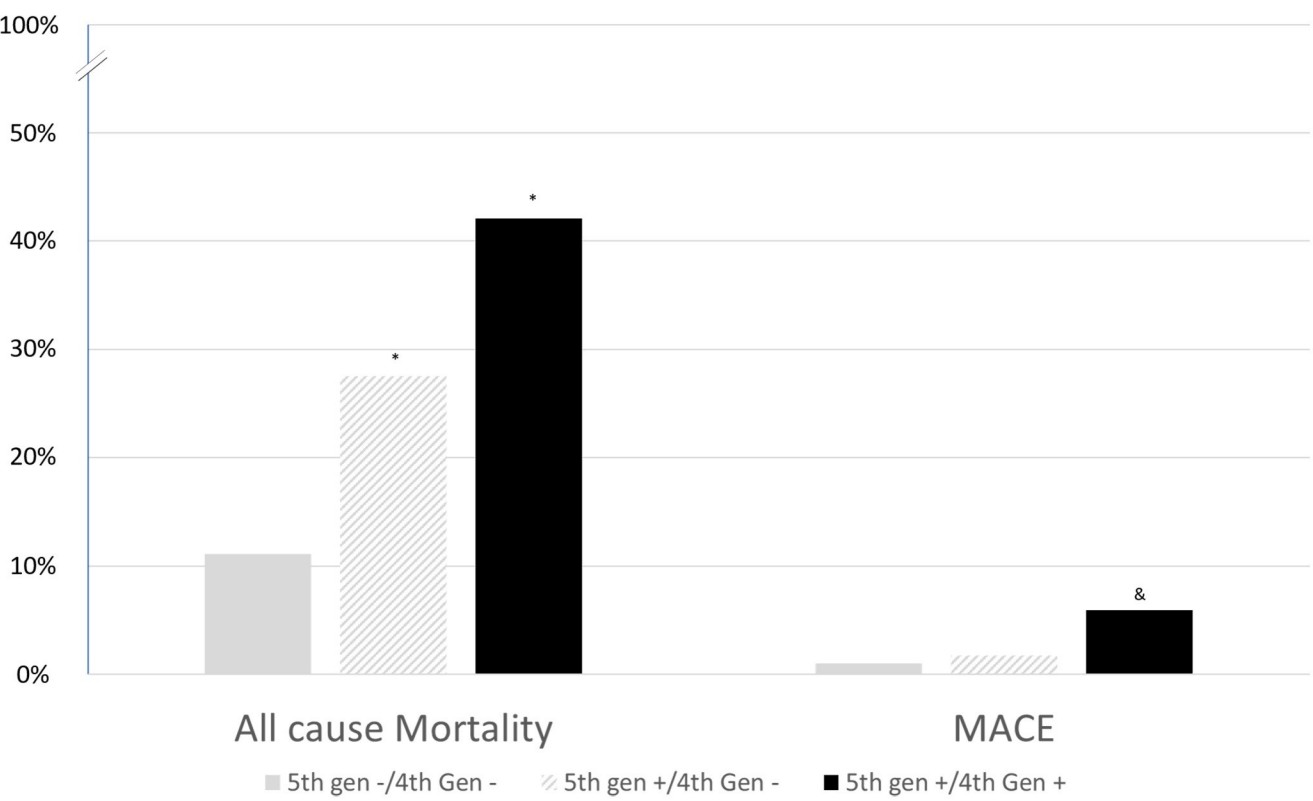

**Fig 3. All-cause mortality and MACE rates based stratified based on 4th generation troponin T and 5th generation troponin T status.** *P-value <0.05 compared to 4th generation—/ 5th generation–for all-cause mortality. &P-value <0.05 compared to 4th generation—/ 5th generation–for MACE.

but significantly lower than some previous reported cohorts [6–8, 11, 13, 14, 19, 20]. Accurately identifying cardiac risk is essential, as it not only influences pre-test probability for subsequent patients, but also influences over-interpretation of diagnostic testing that was never studied for this clinical scenario. Given the rates of positive cardiac troponins, a better understanding of cardiovascular risk and the interpretation of troponin is essential to minimizing inappropriate downstream testing and procedures.

The transition from the 4th generation TnT to the 5th generation TnT has led to an increase in sensitivity at the expense of specificity for both ACM and MACE at 1 year. While not surprising, this shift allows for 2 primary clinical benefits. First, the sensitivity/specificity profile of the 4th generation TnT is difficult to apply clinically, while the improved sensitivity of the 5th generation TnT, one may be considerably less concerned about future MACE in the biomarker negative patient. The decreased specificity does not help in the interpretation of a positive 5th generation TnT (which represents a majority of the patients), but it makes at least one potential result actionable given the improved sensitivity. Second, the 5th generation TnT identifies a cohort of patients at risk for ACM that was previously missed. Amongst those that were 5th generation TnT positive and 4th generation TnT negative (the previously missed cohort), mortality rate was ~25–30% at 1 year, which is lower than the concordantly positive patients, but higher than the concordantly negative patients. This relationship does not hold for identification of MACE. While improving the overall sensitivity, the rates of MACE in the discordant troponin patients were similar to concordantly negative troponin patients, and significantly less than the concordantly positive troponin patients. Given lower than expected MACE and higher sensitivity, the role of the 5th generation TnT is largely in ruling out those at risk for MACE, but of limited value

in the positive test in isolation. Furthermore, the 5th generation TnT may be of added prognostic value for predicting ACM, similar to previous studies, more so than the 4th generation TnT.

However, the limitation in specificity with the 5th generation TnT affects the overall area under a receiver operator curve, making it comparable to the 4th generation in overall accuracy. For both assays, troponin interpretation in isolation is limited in the hospitalized, non-ACS patient population. Previous studies have tried to assess for acute myocardial injury versus acute myocardial infarction in non-ACS presentations, but often were primarily in ED patients that included primary admissions for heart failure and arrhythmias and included a more liberal diagnostic criteria for outcomes (usually ICD codes). In addition to overestimating disease burden, there may also be added issues with introducing bias by identifying a more heterogeneous cohort of both primary cardiac and non-cardiac patients. Given the lower than expected MACE rates, subgroup analysis looking at different classifications would be intrinsically underpowered. This suggests the need for additional strategies for the assessment of cardiovascular risk in hospitalized patients. Ongoing research should help to provide some guidance, whether it be different cutoffs for cardiac troponin, other biomarkers or imaging modalities to help identify a subgroup at risk for MACE [18, 22–27].

## Conclusions

All-cause mortality and cardiovascular risk are elevated in patients hospitalized for non-cardiac presentations compared to the general population. 5th generation TnT is more sensitive, and less specific in predicting both ACM and MACE with limited overall accuracy for predicting both outcomes. However, the 5th generation TnT appears to have an incremental value in identifying an intermediate risk group for all-cause mortality that was previously missed with the 4th generation TnT assay. The 5th generation TnT should be leveraged as another marker of end-organ involvement from a systemic illness, and not necessarily equivalent to or an adequate predictor of major adverse cardiovascular events in isolation. Additional tools need to be identified to use with clinical risk factors and cardiac biomarkers to identify those truly at risk for MACE.

## Supporting information

**S1 Fig. Patient selection flow chart (STROBE Diagram).**
(TIF)

**S2 Fig. Correlation at lower levels between the 4th generation Troponin T assay and the 5th generation Troponin T assay on the same blood samples.** Per sample correlation between 4th and 5th generation Troponin T assays. No patients had a sample that was positive on the 4th generation Troponin T assay but was negative on the 5th generation troponin T assay.
(TIF)

**S3 Fig. All-Cause Mortality (ACM) and MACE rates based on reason for admission.** ACM and MACE rates by reason for admission. No statistical analysis was run due to low event rates.
(TIF)

**S1 Table. Sensitivity, specificity, positive predictive value, negative predictive value and accuracy of 4th and 5th generation troponin T in predicting all-cause mortality.**
(DOCX)

**S2 Table. Sensitivity, specificity, positive predictive value, negative predictive value and accuracy of 4th and 5th generation troponin T in predicting MACE.**
(DOCX)

**S1 Dataset.**
(XLSX)

## Author Contributions

**Conceptualization:** Vedant Gupta, Steve Wah Leung, Vincent Leigh Sorrell.

**Data curation:** Vedant Gupta, Marc Paranzino, Talal Alnabelsi, Karam Ayoub, Joshua Eason, Andin Mullis, Andrew Parks, Levi May, Sethabhisha Nerusu, Daniel Cleland.

**Formal analysis:** Chen Dai.

**Funding acquisition:** Vedant Gupta, Vincent Leigh Sorrell.

**Investigation:** Marc Paranzino, Talal Alnabelsi, Karam Ayoub, Joshua Eason, Andin Mullis, Andrew Parks, Levi May.

**Methodology:** Steve Wah Leung, Vincent Leigh Sorrell.

**Project administration:** Sethabhisha Nerusu, Daniel Cleland.

**Supervision:** Vedant Gupta, John R. Kotter, Steve Wah Leung, Vincent Leigh Sorrell.

**Validation:** John R. Kotter.

**Writing – original draft:** Vedant Gupta, Marc Paranzino.

**Writing – review & editing:** Talal Alnabelsi, Karam Ayoub, Joshua Eason, Andin Mullis, John R. Kotter, Andrew Parks, Levi May, Sethabhisha Nerusu, Chen Dai, Daniel Cleland, Steve Wah Leung, Vincent Leigh Sorrell.

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
