## [Decision Letter · Decision Letter 0]

27 Nov 2020

PONE-D-20-35226

5th generation vs 4th generation troponin T in predicting major adverse cardiovascular events and all-cause mortality in patients hospitalized for non-cardiac indications: a cohort study

PLOS ONE

Dear Dr. Gupta,

Thank you for submitting your manuscript to PLOS ONE. After careful consideration, we feel that it has merit but does not fully meet PLOS ONE’s publication criteria as it currently stands. Therefore, we invite you to submit a revised version of the manuscript that addresses the points raised during the review process.

We look forward to receiving your revised manuscript.

Kind regards,

Giulia Bivona

Academic Editor

PLOS ONE

Additional Editor Comments:

After addressing the Reviewer's issues, the paper can be considered for publication.

Journal Requirements:

2. In your Methods section, please provide additional information about the participant recruitment method and the demographic details of your participants. Please ensure you have provided sufficient details to replicate the analyses such as descriptions of where participants were recruited and where the research took place (location/hospital/site).

3. Please provide a sample size and power calculation in the Methods, or discuss the reasons for not performing one before study initiation.

Reviewers' comments:

Reviewer's Responses to Questions

**Comments to the Author**

1. Is the manuscript technically sound, and do the data support the conclusions?

Reviewer #1: Yes

Reviewer #2: Yes

2. Has the statistical analysis been performed appropriately and rigorously? 

Reviewer #1: Yes

Reviewer #2: Yes

3. Have the authors made all data underlying the findings in their manuscript fully available?

Reviewer #1: Yes

Reviewer #2: Yes

4. Is the manuscript presented in an intelligible fashion and written in standard English?

Reviewer #1: Yes

Reviewer #2: Yes

5. Review Comments to the Author

Reviewer #1: This is a well-written, well-organized and well-illustrated paper. It presents the results of original research and makes a valuable contribution to knowledge and understanding of cardiac biomarkers. In particular, authors compare 5th generation versus 4th generation troponin T in predicting major adverse cardiovascular events and all-cause mortality in patients hospitalized for non-cardiac indications. They found that 5th generation TnT is more sensitive and less specific in predicting both all-cause mortality and major adverse cardiovascular.

I have just a few small comments on the text, which the authors may wish to address:

• I recommend avoiding the use of the abbreviation the first time it was used in the text. After you define an abbreviation (regardless of whether it is in parentheses), use only the abbreviation. Do not alternate between spelling out the term and abbreviating it.

• I suggest to text-align

• According to submission guidelines, do not use non-standard abbreviations unless they appear at least three times in the text.

• I suggest to remove the comma before to cite reference in the text.

• References should be formatted as indicated by the guidelines (i.e. Hou WR, Hou YL, Wu GF, Song Y, Su XL, Sun B, et al. cDNA, genomic sequence cloning and overexpression of ribosomal protein gene L9 (rpL9) of the giant panda (Ailuropoda melanoleuca). Genet Mol Res. 2011;10: 1576-1588).

Finally, a note on the completeness of the bibliography; in order to improve the bibliography I suggest completing the list of references with the following excellent articles: PMID: 28262193, PMID: 21288172, PMID: 19732763.

I recommend publication of this work immediately in PlosOne following consideration of the minor points above.

Reviewer #2: The manuscript is very interesting and well-written. However, I have some minor suggestions to improve its quality.

In the introduction, in the first sentence I suggest to replace preferred with recommended by guidelines.

The main limitation is that the paper does not report seminal studies in this field. Please, consider citing the following: PMID: 29698621, PMID: 31223265, PMID: 29180917

6. PLOS authors have the option to publish the peer review history of their article (what does this mean?). If published, this will include your full peer review and any attached files.

Reviewer #1: No

Reviewer #2: No

---

## [Author Response · Author response to Decision Letter 0]

12 Jan 2021

Response to Reviewers

Editor Comments

Thank you for providing these, and our apologies for not doing this initially. We have modified the Title page and text to conform to these guidelines.

2. In your Methods section, please provide additional information about the participant recruitment method and the demographic details of your participants. Please ensure you have provided sufficient details to replicate the analyses such as descriptions of where participants were recruited and where the research took place (location/hospital/site).

Thank you for your recommendation. We have added some details in the first paragraph of the Methods section (under Study Population) specifically to allow for more reproducibility. 

3. Please provide a sample size and power calculation in the Methods, or discuss the reasons for not performing one before study initiation.

Given the study design, where each participant served as their own control and was included in both the study arm (5th generation TnT) and the control arm (4th generation TnT), there was not a traditional sample size calculation undertaken as the event rate would be the same in both arms. However, based rates of all-cause mortality and major adverse cardiovascular event rates on previous studies done at our institution using a similar definition of outcomes, we predicted 1000 patients would be needed to show adequate events. This has been added under the Methods section (under study population).

There are no restrictions, and we have uploaded a data set. 

Reviewer Comments

Reviewer #1: This is a well-written, well-organized and well-illustrated paper. It presents the results of original research and makes a valuable contribution to knowledge and understanding of cardiac biomarkers. In particular, authors compare 5th generation versus 4th generation troponin T in predicting major adverse cardiovascular events and all-cause mortality in patients hospitalized for non-cardiac indications. They found that 5th generation TnT is more sensitive and less specific in predicting both all-cause mortality and major adverse cardiovascular.

I have just a few small comments on the text, which the authors may wish to address:

• I recommend avoiding the use of the abbreviation the first time it was used in the text. After you define an abbreviation (regardless of whether it is in parentheses), use only the abbreviation. Do not alternate between spelling out the term and abbreviating it.

Thank you for this note. We have reviewed and minimized the use of abbreviations, and only used the abbreviations once defined. 

• I suggest to text-align

We have changed the formatting to justified text align. 

• According to submission guidelines, do not use non-standard abbreviations unless they appear at least three times in the text.

Thank you for pointing this out, and we agree. We have decreased the use of abbreviations only to those terms used repeatedly to simply. 

• I suggest to remove the comma before to cite reference in the text.

Thank you. We have reviewed the comma before in-text citations.

• References should be formatted as indicated by the guidelines (i.e. Hou WR, Hou YL, Wu GF, Song Y, Su XL, Sun B, et al. cDNA, genomic sequence cloning and overexpression of ribosomal protein gene L9 (rpL9) of the giant panda (Ailuropoda melanoleuca). Genet Mol Res. 2011;10: 1576-1588).

We have reviewed the formatting of the references for consistency.

Finally, a note on the completeness of the bibliography; in order to improve the bibliography I suggest completing the list of references with the following excellent articles: PMID: 28262193, PMID: 21288172, PMID: 19732763.

Thank you for directing us to these articles. They have been added.

I recommend publication of this work immediately in PlosOne following consideration of the minor points above.

Reviewer #2: The manuscript is very interesting and well-written. However, I have some minor suggestions to improve its quality.

In the introduction, in the first sentence I suggest to replace preferred with recommended by guidelines.

We agree and have changed the verbiage in paragraph 1 of the introduction. 

The main limitation is that the paper does not report seminal studies in this field. Please, consider citing the following: PMID: 29698621, PMID: 31223265, PMID: 29180917

Thank you for directing us to these articles. They have been added.

---

## [Decision Letter · Decision Letter 1]

19 Jan 2021

5th generation vs 4th generation troponin T in predicting major adverse cardiovascular events and all-cause mortality in patients hospitalized for non-cardiac indications: a cohort study

PONE-D-20-35226R1

Dear Dr. Vedant Gupta,

We’re pleased to inform you that your manuscript has been judged scientifically suitable for publication and will be formally accepted for publication once it meets all outstanding technical requirements.

Kind regards,

Giulia Bivona

Academic Editor

PLOS ONE

Additional Editor Comments (optional):

Reviewers' comments:

Reviewer's Responses to Questions

**Comments to the Author**

1. If the authors have adequately addressed your comments raised in a previous round of review and you feel that this manuscript is now acceptable for publication, you may indicate that here to bypass the “Comments to the Author” section, enter your conflict of interest statement in the “Confidential to Editor” section, and submit your "Accept" recommendation.

Reviewer #1: All comments have been addressed

Reviewer #2: All comments have been addressed

2. Is the manuscript technically sound, and do the data support the conclusions?

Reviewer #1: Yes

Reviewer #2: Yes

3. Has the statistical analysis been performed appropriately and rigorously? 

Reviewer #1: Yes

Reviewer #2: Yes

4. Have the authors made all data underlying the findings in their manuscript fully available?

Reviewer #1: Yes

Reviewer #2: Yes

5. Is the manuscript presented in an intelligible fashion and written in standard English?

Reviewer #1: Yes

Reviewer #2: Yes

6. Review Comments to the Author

Reviewer #1: (No Response)

Reviewer #2: (No Response)

7. PLOS authors have the option to publish the peer review history of their article (what does this mean?). If published, this will include your full peer review and any attached files.

Reviewer #1: No

Reviewer #2: No

---

## [Editor Report · Acceptance letter]

28 Jan 2021

PONE-D-20-35226R1 

5^th^ generation vs 4^th^ generation troponin T in predicting major adverse cardiovascular events and all-cause mortality in patients hospitalized for non-cardiac indications: a cohort study 

Dear Dr. Gupta:

I'm pleased to inform you that your manuscript has been deemed suitable for publication in PLOS ONE. Congratulations! Your manuscript is now with our production department. 

Kind regards, 

on behalf of

Dr. Giulia Bivona 

Academic Editor

PLOS ONE